# Recent Progress in 3D Printed Mold-Based Sensors

**DOI:** 10.3390/s20030703

**Published:** 2020-01-28

**Authors:** Shan He, Shilun Feng, Anindya Nag, Nasrin Afsarimanesh, Tao Han, Subhas Chandra Mukhopadhyay

**Affiliations:** 1School of Chemistry and Chemical Engineering, Guangzhou University, Guangzhou 510006, China; shan.he@gzhu.edu.cn; 2Institute for NanoScale Scale Science and Technology, College of Science and Engineering, Flinders University, Bedford Park 5042, Australia; 3School of Electrical and Electronic Engineering, Nanyang Technological University, Singapore 639798, Singapore; shilun.feng@ntu.edu.sg; 4DGUT-CNAM Institute, Dongguan University of Technology, Dongguan 523808, China; hant@dgut.edu.cn; 5School of Engineering, Macquarie University, Sydney 2109, Australia; subhas.mukhopadhyay@mq.edu.au

**Keywords:** 3D printing, mold, sensors, casting, microcontact

## Abstract

The paper presents a review of some of the significant research done on 3D printed mold-based sensors performed in recent times. The utilization of the master molds to fabricate the different parts of the sensing prototypes have been followed for quite some time due to certain distinct advantages. Some of them are easy template preparation, easy customization of the developed products, quick fabrication, and minimized electronic waste. The paper explains the different kinds of sensors and actuators that have been developed using this technique, based on their varied structural dimensions, processed raw materials, designing, and product testing. These differences in the attributes were based on their individualistic application. Furthermore, some of the challenges related to the existing sensors and their possible respective solutions have also been mentioned in the paper. Finally, a market survey has been provided, stating the estimated increase in the annual growth of 3D printed sensors. It also states the type of 3D printing that has been preferred over the years, along with the range of sensors, and their related applications.

## 1. Introduction

Since the beginning of the usage of sensors for ubiquitous monitoring [1,2], researchers have been trying to optimize the techniques for their fabrication. At around three decades ago from now, MEMS-based sensors were popularised where the substrates of those prototypes were mainly silicon [3,4]. While these semiconducting sensors have been applicable for different sectors like environmental and industrial applications, there were some constraints attached to these sensors which needed immediate alteration. Some of them were their high cost of fabrication process which was due to the technologically advanced equipment, requirement of a long time to fabricate a single sensor due to the involvement of several steps of fabrication, incontinency caused on biomedical applications, high input power, degradation of their sensitivity, and saturation of their response after prolonged usage. Among these constraints, the technique of fabrication has the most diverse effect as it involves both time and money. Additionally, the development of silicon-based sensors required expertise, starting from the first step of forming single-crystal silicon to the sputtering step to form the electrodes [5,6]. One small error in the fabrication requires the fabrication process to be repeated. In alteration to the silicon sensors, flexible sensing prototypes [7,8] were developed that increased the application domain of the sensors in real-time.

Researchers have been working on the various types of fabrication processes of flexible sensors for a while [9,10,11] to optimize the time and cost for each prototype, without compromising their quality. Among the different types that have been used, printing methods [12,13,14] have always fascinated the researchers due to the advantages associated with them. Some of them are advanced mechanical attributes like thinness, lightweight, flexibility, electrical enhancement, roll-to-roll production, and versatility of the raw materials that can be processed to form the sensors. Some of the popular printing techniques are flexography [15,16], inkjet printing [17,18], screen printing [19,20], offset lithography [20,21], and 3D printing [22,23]. Out of these methods, 3D printing technique has been highly favorable as it has substantial advantages over other techniques like faster large scale production of the prototypes, easy customization of the sensor structure, reduced cost of fabrication, risk mitigation, reduced complexity of the fabrication, and sustainability [24,25]. A 3D printing technique has been employed by a large number of researchers to design, fabricate, and implement their respective sensing prototypes for a range of sectors including electrochemical [26,27], strain [28,29], and electrical [30,31] sensing applications.

The work on 3D printed molds has been done a while to exploit the above-mentioned advantages for the fabrication of flexible sensing systems. Even though review works [23,32,33,34] show the detailed work of the use of 3D printing to form different kinds of prototypes, there are some drawbacks that need to addressed through the current paper. Among the various types of 3D printing methods, none of the review papers distinguishes the relationship between the types of 3D printing and their advantages in relation to their associated applications. Secondly, even though a general and unspecific view of the sensors for different types of applications has been mentioned in these papers, a categorization in the applications for particularly the mold-based sensors has not been highlighted. This paper addresses both these points as it explicitly mentions the use of the 3D printed mold-based to form the sensors, particularly for biomedical applications. This paper also classifies the types of biomedical sensing, based on the type of fabricated prototype. The fabrication of different types of molds has been showcased in the following section, where each of them has been developed with different materials, and subsequently employed to form sensors for biomedical applications. This section sub-categorizes the types of biomedical applications each of these prototypes have been employed for. Table 1 shows a comparative study of some of the significant research works done on 3D printed mold-based sensors. The study has been done based on the type of materials used to fabricated the sensors, the advantages, and applications associated with them.

The rest of the manuscript is organized as follows. Followed by the brief introduction given in Section 1 explaining the need for 3D printed mold-based sensors, the fabrication procedure of a range of sensing prototypes and their applications has been explained in Section 2. Section 3 explains some of the challenges remaining with the current sensors, as well as some of the possible remedial solutions. A summarization has also been provided to highlight the advantages related to the range of processed materials in this type of 3D printing, and their specific use for the proposed application. A market survey is also given depicting the use of 3D printed sensors in recent years, along with an estimation of their uses in future years. The conclusion is provided in the final section of the paper.

## 2. 3D Printed Mold-Based Sensors for Different Applications

This section includes some of the mold-based devices that were printed and cast upon to form sensing prototypes. The fabrication and employment of the sensors for different applications have been explained here. The research works have been categorized on the basis of biomedical and industrial uses. Some of the molds were developed using different kinds of computer-aided design (CAD) models to obtain different kinds of molds for sensing prototypes [44,45,46]. In many cases, the 3D printing technique has been utilized directly on the substrates to develop smart objects. The endurance and dimensions of the developed prototypes depended on the quality and resolution of the printing device.

### 2.1. Biomedical Applications

One of the interesting works is related to the usage of 3D printed molds for biomedical uses is the development of capacitive pressure sensors via the casting of polydimethylsiloxane (PDMS) on micro-structures [42]. The size of each mold was 3 × 3 cm, with each one containing periodical micro-grooves. The molds were formed using acrylonitrile butadiene styrene (ABS) as the printing filament where the depth and width of the micro-grooves were 100 and 150 microns, respectively. Followed by the casting and curing of free-standing PDMS film, lamination of indium-tin-oxide (ITO) coated poly (ethylene terephthalate) film was coated on the PDMS and subsequently cut into small pieces. After two pieces of each of these laminated structures were attached face-to-face with the PDMS films, the devices were made electrically conductive using copper tapes on the ITO electrodes. The relative change in capacitance of these structures was measured as a function of the pressure applied to the sensors. Different values of sensitivities of 1.62 and 0.05 kPa^−1^ were obtained for low pressure and high-pressure ranges, respectively. The sensors were used for monitoring human physiological signals by attaching the sensors on the radial artery of the wrist of a human. The sensors were also tested with a rice grain with a weight of 15 mg to determine their capability of determining very low-pressure. The sensors depicted a fast response and recovery time with excellent durability to ambient temperature and humidity. Another interesting work related to the 3D printed mold-based sensor is the fabrication of capacitive sensors based on micro-structured PDMS film that was used as a dielectric layer [47]. The sensors were used as tactile sensing implementing their stretchable nature. Some of the applications of the sensors are related to their employment as electronic skins via using them as prosthetic skins. Flexible, capacitive sensors were fabricated over a large area by forming microstructures on thin-films developed on PDMS. The developed sensors had a tunable sensitivity, which can be altered by using different microstructures. The electrodes were developed by mixing the carbon nanomaterials and metal nanoparticles by obtaining a percolation threshold between them to obtain optimized electrical conductivity. The integration of the formed microstructures with the organic field-effect transistors generated excellent sensitivity and response time. Figure 1 [47] shows the schematic diagram of the fabrication of the microstructure PDMS films. The PDMS mixture formed by mixing the pre-polymer and cross-linker was drop cast onto a mold formed with silicon. The molds were formed with wafers having a thermally grown oxide layer of 300 nm thickness. The patterning on the molds was done using the photolithographic technique, with a mask of potassium hydroxide for etching purposes. The mold contained the arrays of the inverse of the features that were to be developed. This mold was laminated by an ITO-coated PET substrate and subsequently cured at 70 °C for three hours. The thickness of the ITO-contained PET film was around 150 microns, which was treated with ultraviolet ozone for twenty minutes. Finally, the flexible substrate was peeled off the mold.

Other notable works include the employment of electronic 3D printing techniques [22] and facile methods [48] to develop strain sensing and actuators. These devices were highly programmable and have been used to develop customized prototypes in a seamless manner. These strain sensors have been able to be deformed by more than 20% with a very high gauge factor of 50,000. These prototypes have been employed along with a glove to determine the kinematics of a finger. The irregularity of the movement of the finger patterns has been assessed in real-time at different stiffness or dexterity. Another similar work [49] involves the use of polymer and ionic liquid composites to analyze the effect of crosslinking and polymerization of printed composite in terms of piezoresistive sensing elements. The ionic liquid and polymer used to form the composites were (1-ethyl-3-methyl-imidazolium tetrafluoroborate (EMIMBF_4_) and 2-[[(butyl amino) carbonyl] oxy] ethyl acrylate (BACOEA)) respectively. The synergy of the ionic transport properties of EMIMBF_4_ was improved by the coordinate sites developed by the location motion of BACOEA chain segments. This synergic action took place under enough activation energy to enhance the phenomenon. The resulting strain on the sensors reduced the electrical resistance due to the increase in the resultant distance between the electrodes, which as a result, led to the motion of polymer chains and IL. One of the advantages of this process is the photo-curability of all the materials that been used to develop the stretchable, piezoresistive tactile sensors.

Another interesting work is related to the formation of bioinspired cilia sensors using 3D printing and casting techniques [50]. The technique involved 3D printing of a metallic mold containing complex features such as micropillar and microchannel, where PDMS and graphene nanoplatelets were cast and drop-cast respectively to form the desired features. The finished prototype was piezoresistive in nature, forming a flexible strain gauge. A very high gauge factor of 37 was obtained from these sensors via cyclical tension-compression tests. The cilia like nature of the pillars, along with a cantilever and microchannel, were responsible for showing good sensitivity towards tactile and flow sensing. The limit of detection for the tactile and flow sensing was 12 microns and 58 mm/s respectively. A layer-by-layer technique was used to form the molds with each layer having a thickness of 30 microns. The molds were fabricated containing a cantilever-pillar design with a dimension of 1.5 × 4 mm of each of the cylindrical hair-like structures, that was located at the free end of the horizontal cantilever. The presence of an additional U-shaped microchannel having a dimension of 0.3 × 0.3 × 6.5 mm near the top surface of the cantilever increased the sensing element of the prototype. The processing parameters of the laser used for developing the sensors were 200 W, 800 mm/s, and 120 microns as power, speed, and hatch spacing respectively. The printing of the molds was followed by sandblasted and lubricated using a commercial degreaser, to improve their surface quality. The final steps included the casting of PDMS, desiccating the molds for around 40 min and curing to obtain the structure. Finally, a conductive dispersion containing graphene, n-butyl acetate and ethanol was drop cast on the microchannel of the cantilever surface. Figure 2 [50] shows the schematic diagram of the fabrication process of the 3D printing and casting process. The fabricated prototypes showed an average gauge factor and sensitivity of 37 ± 1.7 and 30 mV/(ms^−1^). The sensors were tested for respiratory exhalation and oscillatory flow stimuli in deionized water to determine their capability as strain and flow sensors, respectively.

One of the exciting works done by Nag et al. was based on the use of graphite and PDMS on 3D printed molds to develop sensors for biomedical [35,51] and environmental [52] applications. Figure 3 [35] shows the schematic diagram of the fabrication process. The difference of this procedure from the other ones lies in the use of 3D printing and casting techniques to develop the sensor patches. ABS with a diameter of 1.75 mm was used as the printing filament to produce the molds due to its natural processing high-temperature resistance. Followed by the fabrication of the patterns, graphite powder was cast on the trenches of the molds to form the electrodes of the sensor patches. The amount of graphite powder was optimized by varying the channels of the 3D printed molds. The trading off was done between the electrical conductivity and flexibility of the sensor patches. The optimized height of the trenches was around 500 microns. Then, PDMS was cast on top of the molds whose height was adjusted to 1000 microns to define the substrate of the sensor patches. The samples were then desiccated and cured at 70 °C for 2 h to solidify the samples. The sensor patches were then peeled off and used for characterization and experimentation purposes [51]. The total sensing area was 900 mm^2^ constituting three pairs of interdigitated electrode fingers. Each finger had a length and width of 18 and 2 mm, respectively. The advantages of these sensor patches lied in its easy, cheap and quick fabrication process and formation of conductive nanocomposite-based sensors. In terms of biomedical applications, these prototypes have been used as strain sensors for determining movements of different parts of the body. The sensors were attached to various strain-induced locations like elbow, knee, neck, and joint of a finger to determine their motion. Oscillatory motions of the body parts were performed to determine the repeatability in the responses. The sensors were successful in determining the small-scale strains induced on the body, which are necessary for rehabilitation purposes. These sensors were also used for low-force sensing, where very low pressure was exerted on the sensing area of the patch to determine the change in the response of the prototypes. The sensors were able to measures forces ranging between 3.5 to 17.5 mN with a sensitivity of 0.25 pF/mN. The measurements were taken in terms of change in capacitance with respect to the input force values. These experiments served a purpose of touch sensing of real-time objects to determine the degree of force patients having artificial skins. The environmental application included the measurement of phosphate concentrations via development of an embedded microcontroller-based system. The development prototypes were able to determine phosphate concentrations ranging between 0.03 and 30 ppm, which is present in the natural water bodies. The sensor output was processed through a microcontroller to the monitoring unit for further analysis. The responses were repeatable in nature, having a standard deviation and sensitivity of 3% and 0.0498 Ω/ppm. These applications served can be highlighted to use this sensor as a multifunctional sensing prototype.

Liu et al. [53] also designed a conductivity sensor that collects sweat from the skin by a PDMS based sweat collector, made by 3D printed plastic molds. This wearable sensing system monitors the sweat in real-time like a wristwatch from the conducted value of the collected sweat from the forearm and sends the data wirelessly. Heger et al. [54] had designed and manufactured a 3D printed immunosensor, with biocompatible PDMS chip used for magnetic isolation, and quantum dots-based immune labeling of metallothionein as the biomarker for various types of cancer, including spino-cellular carcinoma. Polymer acrylonitrile butadiene styrene is used as a 3D printed material due to its better optical transparency, critical surface tension, stiffness, and chemical resistance. Another interesting work explained by Hwang et al. [55] was based on the development of PDMS-based balloon fin actuators that reduces stress concentrations. The use of 3D printed molds indicates the development of pneumatic micro-fingers using balloon fins to obtain a linear motion. Figure 4 [55] shows the photograph of the fabricated balloon fin actuators formed with different geometries. The design and implementation were done to generate vertical micro-balloon fins that were capable of generating a linearized change in the bend angle with respect to the application of pressurized air. The designs of the molds were done using CAD files, which were fed into a 3D printing device that would simultaneously jet photo-curable polymer and sacrificial multilayer with a thickness of 28 microns. The completion of the printing of molds was exposed to a high-pressure water jet from the printing device to dissolve the supporting material. PDMS formed with a mixture of 15:1 ratio between the pre-polymer and curing agent, was poured on the molds, desiccated, and cured at 75 °C for 2 h to solidify the substrate. The balloon cavities were formed after placing the molds on a 3D printed holder that maintained asymmetry on both sides of the cavities for deflection. The printed molds were coated with Au-Pd as the passivation layer from different angles to prevent the inhibition of the thermal cross-linking effect of the PDMS pre-polymer on the native 3D printed material. The roughness of the printed molds hurt the passivation layer. The PDMS was then taken off the molds to form the balloon actuator where thin cast shells with a thickness of 100 microns were formed to obtain pliant and conductive shell structures. A comparison was made between all the shell structures that were developed using the 3D printed molding process. The width of the balloon fins was fixed at around 100 microns to implement the linear change. The 3D printed molds for this application had an advantage of additional freedom in terms of the shape of the actuators, in comparison to the extruded 2D shapes. The experiments were done to determine the exceeding bend angle for static bending and mechanical force of the fabricated micro-fingers with different kinds of analytical models. The micro-fingers were successfully able to determine the complex object transfer tasks. These balloon fins in the presence of micro-fingers were able to reduce the stress concentrations that are highly probable for 2D structures to cause material failure.

A 3D flexible tactile sensor was developed by Lee et al. [56] by integrating a 3D printed PDMS molding with a nanofiber mat. This wearable device can be used to measure the tactile force and even skin deformation caused by the tester’s pulse. The reaction forces of the fingertip against pressing an object can also be used in healthcare devices and soft robotics. After the printing of the molds, the PDMS substrate was cured for 4 h at 80 °C and integrated with PVDF-TrFE nanofibrous mats. He et al. [57] fabricated soft tissue prostheses mold by medical grade silicone using a desktop 3D printer. Since silicone provides extra strength and elasticity, these soft prostheses such as ear and nose can be easily pulled out from the mold. The ABS copolymers based filament was used as a printing material in the 3D printing process. After printing of the sample of eight 0.2 mm layer, it was allowed to dry out for about 1 h. One of the advantages is that this silicone prosthesis possesses high tensile strength, tear strength, and good heat resistance above 200 °C. Figure 5 [57] shows the individual diagram of the molding and demolding processes.

Russo et al. [58] introduced a soft and stretchable silicon-based sensor that can be stretched to deform the cross-section for measuring the change in electrical resistance. A biocompatible NaCl solution-filled channel and gold stretchable electrodes were integrated with this silicon rubber. This electrode-based sensor can be used in medical applications by measuring the applied strain during large deformations. The silicon rubber-based 3D printed mold was kept for four hours at room temperature and then cured. One of the interesting works [59] is associated with the fabrication of metal–metal oxide nanostructures on freestanding graphene that have the potential to be employed for bio-sensing, bioelectronics, and lab-on-a-chip devices. Figure 6 [59] shows the schematic diagram of the fabrication process of the paper-based biosensing platform. High-density platinum nanoparticles were deposited on freestanding reduced graphene oxide paper that carried MnO_2_ networks. The fabrication of the graphene paper was done via evaporation and chemical reduction of an aqueous dispersion of exfoliated graphene oxide nano-sheets that are cast on printed molds. The template-free electrodeposition of MnO_2_ nanowire networks was done on the reduced graphene oxide substrates to induce a larger surface area via 3D systems. Graphene oxide was synthesized using modified Hummer’s method and formed an aqueous solution to cast on polytetrafluoroethylene-based mold. The thickness of the graphene oxide paper was varied between 1 to 30 microns by varying the volumes of graphene oxide dispersion. The graphene oxide solutions were then reduced by immersing them in an Hydrogen Iodide (HI) solution at room temperature for an hour. The reduced graphene oxide was then thoroughly washed with deionized water and dried at room temperature for 10 h. The electrodeposition and ultra-sonication of platinum nanoparticles done on the MnO_2_ network-coated graphene paper were based on the controlled size and morphologies of platinum nanoparticles that were fabricated using wet-chemical and electrochemical approaches. Cyclic voltammetry was done on the reduced graphene oxide electrodes after immersing them into NaSO_4_ and MnSO_4_. The obtained papers were then used for potentiostat deposition of platinum nanoparticles via ultrasonic irradiation. The samples were then again washed, dried, and stored in the vacuum desiccator. The effect of the proper size distribution of the nanoparticles achieved good versatility in catalytic and sensing processes. The advantage of this approach was related to avoiding size limitations, which increased the flexibility of scaling-up of the sizes in different applications.

Wang et al. [60] developed a flexible and highly sensitive tactile sensor that can be used in biomonitoring electronic devices and robotic parts. It has an interlocked truncated saw-tooth structure, which converts the external pressure to the tensile strains. The sensor was composed of a layer-based 4 × 4 sensing unit with a resolution of 5 mm and the top and bottom layers were fabricated by mold casting. Liquid Silicon Rubber was used to fabricate the layers by pouring it into the plastic mold and heat was applied to cure the Silicone Rubber (SR). Due to this SR, the top and bottom layer and graphene layer exhibit high flexibility. Also, the flexible printed circuit was made by row and column electrodes. Figure 7 [60] shows the overview of the different layers of the sensing array, their electrode configuration and the cross-sectional view of the prototype that was used pressure detection.

Truby et al. [61] fabricated a 3D printed soft robotic gripper having a multi-degree of freedom soft actuator that can be used as a tactile sensor. The 3D printed fingers are used to tip, base, and full finger actuation. Organic ion gel comprised of 1- ethyl-3-methyl-imidazolium ethyl sulfate (Sigma) filled with 6 wt.% fumed silica particles (Aerosil 380, Evonik) were used as printing ink and it helps to impart the appropriate yield stress needed for printing. It takes 90 min to print the finger and 12 h to cure the ink. This researcher group also developed an algorithm to hold the object using its unique actuation mode. Lee et al. [56] fabricated a flexible tactile sensor using piezoelectric nano-fiber using 3D printed mold assembly. A core piezoelectric layer and a lower and upper sandwiching substrate were used to fabricate this mechanical sensor. This sensor is capable of sensing the scanning data of the fingertip and wrist as the human interface device. Also, it can measure the forces generated during the holding of the object by the fingertip and can monitor the pulse waveform. In the field of health monitoring and soft robotics, this sensor has wide applications. The mold assembly was printed in a single scan without any assembly procedure. Due to its adaptive size, this 3D printing technology has a great advantage in developing this structure.

Another research group, [62], also developed a soft robotic gripper that can hold a haptic object as well as twist it. Silicon elastomer was used as the printing material to fabricate the finger mold. A flexible sensory skin was also developed to measure the deformation and contact of the object. Tee et al. [63] developed a tunable sensor to monitor the blood pulse and sense force on the trackpad. First, the research group used Finite Element Method (FEM) to study the compressibility of the PDMS structure and then applied the films between the parallel plate, which will act as a pressure sensor. This low-cost capacitive pressure sensing system has a great advantage of having high mechanical sensitivity. For the larger sensor area, the process of molding manufacturing could be varied.

### 2.2. Industrial Applications

A few of the significant works related to the design and fabrication of 3D printed molds for industrial applications have been mentioned here. One of the interesting works in terms of the use of carbon materials on 3D printed mold-based sensors was done by Hu et al. [64], where flexible tactile sensors were formed on polymeric substrates. Carbon nanotubes (CNTs) were used as the conductive material to form the electrodes by integrating it with flexible printed circuits connectors. The sensors were used for determining normal and shear forces. A 3D distribution of the polymer in the shape of tactile bumped structures was initially done using anisotropically etched silicon molds. This was followed by the planar and 3D distribution of CNTs to form the sensors which were piezoresistive in nature. The patterning of these CNTs, along with the metal routing of FPC connectors, was done using a batch fabrication technique where the rigid silicon was used as the substrate instead of flexible polymers. The electrical connections were further improved using parylene encapsulation in order to avoid delamination. Finally, the entire structure was transferred on the PDMS substrate to form an integrated device. These sensors were used as tactile sensors to conduct experiments on normal and shear forces. Similar work as above was done by Han et al. [65] to develop high sensitivity strain gauges using two different techniques, namely micro-molding and stamping. PDMS and carbon nanoparticles were used as the polymer and conductive material respectively. The design and thickness of the sensors were optimized to improve the piezoresistive coefficient and sensitivity of the prototypes. A prior simulation was done using: Analysis of Systems (ANSYS) to determine the gauge pattern of the sensors. Five different grids for the strain gauge were formed to obtain the highest gauge factor of 140 microns. A nanocomposite was formed using carbon particles and PDMS, along with the use of heptane as the solvent. An optimized value of 5 wt.% of carbon particles was mixed with the elastomer and solvent. The mixture was homogenized by volatilizing the solvent into the Carbon-PDMS-mixed nanocomposite. Prototypes with two to five grids were developed as shown in Figure 8a [65]. SU8 was used as the photoresist during the formation of the molds. The exposure time, development time, and rotation speed during the mold formation were 4.5 s, 9 min, and 1500 rpm respectively. Initially, the casting of PDMS was done using a silicon mold, followed by degassing and curing it. This was used as a stamp and pressed against the silicon molds to transfer the carbon and PDMS molds. Figure 8b [65] shows the optical microscopic image of the mold used to develop the sensor. The sensing structure was completed by forming a sandwich structure of the nanocomposite between two PDMS layers, subsequent to the connection of the nanocomposite layer to conductive layers for electrical measurements. The electrode pads of the strain sensor were connected with aluminum wires using silver glue for subsequently performing piezoresistive sensing experiments. The density, Poisson’s ratio, and Young’s modulus of the developed sensors were 956.0 kg/m^3^, 0.3, and 410 kPa respectively. The tensile stress and strain on the right side of the sensing pattern while fixing the left side were 0.2415 MPa and 25% respectively. The durability of the sensors was tested by determining the change in their relative resistances over a number of cycles. The sensors were capable of lasting for more than sixty cycles; however, the recovery of the shape of the sensors from the tensile stress reduced with the increase in the number of cycles.

Another interesting work related to the use of a 3D printed mold-based sensor was related to the fabrication of the cylindrical arch spring-shaped tactile sensors using conductive polymer [66]. The molding process was used to develop the sensors where a polymer formed by silica and carbon black, coupled with Silane-69, and pressed onto the molds. The cylindrical arch spring structure was considered due to its assistance towards the force scaling purposes for developing customized tactile sensors. The sensor had a capability of determining a force of 90 N, having a maximum structural deflection of 4 mm. The data collected was transferred using a microcontroller-based system that was embedded in the sensor. Figure 9 [66] shows the schematic diagram of the view of the developed sensor. Initially, the cylindrical arch spring structure was developed with the help of simulation using Finite Element Analysis Tools of COMSOL Multiphysics. Three different printing filaments, namely ABSplus, ABS, and PLA with failure stresses of 36, 22, and 26 MPa respectively, were considered for 3D printing of the sensing structure. Among them, due to the minimum stress to the tensile modulus index value of ABSplus, it was chosen to customize the stiffness and strength of the sensing structure. Due to the optimized conductive nature of ECP-CB-1 carbon black at room temperature and pressure, it was mixed with the coupling agent Si-69 and nanopowder silicon dioxide. The polymer was prepared by mixing an optimized value of 8 wt.% of superconductive carbon back with RTV silicone rubber. The mixture was put to a pressing mold to form the sensing element. Three different layers of the molds were formed using the mechanical press in order to ease the ejection process. Finally, the electrodes were formed using copper-clad foils along with a layer of silver paste layer. The printed elements were assembled polymer pill to form a 1-Degree of Freedom tactile sensor. The drift characteristics were observed for 27 min for two different loadings of 5 and 10 kg. The average stiffness and standard deviation of the sensors were 7701.12 and 196.77 N/m respectively. The sensor showed a decrease in the output voltage for the loading and unloading of different weights.

Similar work was presented by Vogt et al. [67] were 3D printed molds were designed and developed to form a force sensor having embedded microfluidic channels. The sensors were thin and highly compliant in nature, having a modulus of 69 kPa. The channel configurations were 200 × 200 microns and 300 × 700 microns for two different types of designed prototypes. A sensitivity of −28.6 mV/N was obtained for two principal in-plane axes. Three different axes were developed along with three channels being separated by 120°. These channels had three sensing elements on each of them to increase the accuracy of the channels. eGaIn was injected into each of the channels, which subsequently combines into to facilitate filling. The channels were then divided into three major channels, with each of them having eight sub-channels, thus having a total of twenty-four channels. The microchannels were poured with an elastomer to form a top and a flat bottom layer. Then, these layers were bonded by a spin coating, which is followed by injecting eGaIn and air in each of the layers. Finally, the electrical connections were made by using wires in the needle holes of each reservoir. The length of the microchannels was kept very short due to their formation with eGaIn with copper traces being used to form the flex circuits. The areas between the eGaIn and copper were tapped with copper traces in order to protect the conductive interface using the spin-coating of a thin layer of silicone adhesive. The in-plane sensitivities were 37 and −28.6 mV/N respectively. The sensitivities to the normal forces were 327.8 and 572.7 mV/N for forces below 3.4 and over 4.1 N respectively. Another similar way [68] to print elastic conductors along with the use of SWCNTs as conductive material was done using the jet-milling technique. The resultant prototypes were rubber-like, stretchable in nature with more than 100% stretchability obtained from the active-matrix display.

One of the interesting works in this area involves the use of metallic micro-structured molds to develop plastic microfluidic devices [69]. The 3D printing technique used to develop these metallic micro-structures molds was a selective laser sintering method. The thermoplastic microfluidic chips were developed using a selective sintering method via the injection molding technique. A cyclic olefin copolymer was used for the injection purpose into the microfluidic chips, which was eventually used to monitor the presence of different concentrations of nitrate in the water. One of the biggest advantages of this technique is the quick fabrication of the molds to develop microfluidic devices for injection purposes. The molds were printed with a power and layer thickness of 100 W and 25 microns respectively. The micro scaling was done the pre-finished substrate via highly-energized fiber laser melting and subsequently cooling the metallic powders. The molds consisted of a series of patterns having rectangular ridges with widths and heights varying from 0.1 mm to 1 and 0.3 mm to 1 mm, respectively. Figure 10 [69] shows the micro ridge arrays along with their enlarged view and printed patterns. Electrical discharge machining was used to pre-machine the substrate and was subsequently positioned in the powder platform for printing purposes. The printing process of the patterns was completed by sing Die-sinking electrical discharge. In order to deal with the roughness of the surface caused by Selective Laser Melting (SLM), electro-polishing was done using an electrolyte composite of 35% of sulphuric acid, 45% of phosphoric acid, and 20% of water. Other advantages provided by this technique were cost-effectiveness, flexibility in the customization of the design of the device, and high quality in the finishing of the prototypes.

Another research group, Vatani et al. [70], also developed a multi-layered tactile sensor using a direct printing technique. An elastomeric material was used to develop the sensing material, which has good flexibility and conformability, as well as mechanical durability and compliance. This tactile sensor helps to determine the forces that are applied over any surface by monitoring the resistivity changes using the sensing tip. This two-layered sensor has 64 distinct positions of taxels which have the advantage of detecting 16 signals from the two layers of sensing elements. The good resolution of the sensor helps to detect the slip which is very important in robotic manipulation. A novel 3D printed tactile sensor [71,72] with a conductive polymer sensing element was introduced by Silva et al. [73]. The sensor comprised cylindrical arch springs, which were in a combination to achieve favorite mechanical properties. The embedded electrical circuitry is used for output signal conditioning. The reported structure was designed and fabricated by 3D printing. One of the works [74] showcasing the use of micro-contact printing technique based on 3D printed molds was related to the formation of nanocomposites for electrodes or sensing probe applications. Multi-walled carbon nanotubes (MWCNTs) and PDMS were mixed at defined proportions and cast on printed molds to form elastomeric devices. The developed devices exhibited excellent mechanical flexibility and electrical conductivity for different loading percentages of Multi-Walled Carbon Nanotubes (MWCNTs). Followed by the formation of the printed molds using micromachining, spin coating of conductive nanocomposites was done to create a thin-film layer. Then, the conductive layer was then transferred from the patterns and stamped on another substrate. This was followed by partial curing of the transferred substrates and subsequently adding more PDMS. The final step includes the curing of the entire PDMS block and de-bonding it from the second temperature to develop the elastomeric device. The pre-polymer of the PDMS was initially mixed with toluene at a ratio of 1:4 in addition to the mixing of MWCNTs with toluene at a ratio of 1:20 in a separate container. The mixing of PDMS and MWCNTs in separate containers were done in volume and weight-based ratios respectively. The MWCNTs–toluene mixture was then magnetically stirred for 2 h and mixing with the PDMS-toluene mixture in the third container. The final mixture was then heated at 50 °C overnight on a hot plate to evaporate the volatile toluene to leave the nanocomposite mixture. Then, the curing agent was added to the mixture at a ratio of 1:10 to make the composite ready for printing purposes. The spin-coating of the nanocomposites was done on a silicon wafer having a thickness of 50 microns. The printing mold was developed with poly (methyl methacrylate) (PMMA) and placed above the spin-coated silicon wafer. The mold was dipped onto the conductive composites and stamped onto a chlorotrimethylsilane-treated glass substrate. The sample was then further heated at 60 °C for half an hour to solidify them further and minimize the chances of diffusion of PDMS onto it in the subsequent steps. Then additional PDMS was poured on top of the composites and cured at 60 °C for 4 h to solidify the substrate. Finally, the sample was de-bonded from the PMMA template to form the sensor.

One of the significant uses of 3D printed molds in forming microfluidic channels was explained by Hwang et al. [75], which showed the formation of microfluidic networks in monolithic PDMS. The micro-channeled molds were 3D printed using CAD files, which obtained molds with a dimension of 50 ∗ 50 ∗ 5 mm. A photocurable polymer, along with a support material of 28 microns, was printed in 30 min. The printed molds were UV-cured and washed with a high-pressure water jet to make them ready for casting PDMS. After the casting of PDMS was done after mixing the pre-polymer and curing agent at a ratio of 9:1, the sample was desiccated, cured at 75 °C, and removed from the molds to form the final device. The devices were then characterized regarding surface roughness and resolution. The minimum dimensions of the microfluidic channels were around 100 s of microns, which makes this technique a better alternative to the soft lithography process. Figure 11 [75] shows the fabrication and extraction process of the 3D-printed mold.

Emon et al. [76] presented the fabrication of piezoresistive sensors that can be embedded in 3D printed tires. The sensors were based on composites developed from CNTs and polymer. The stretchable electrodes were formed by mixing five wt.% of MWCNTs in the polymer to form the composites. Triton X100 and DMF were added as surfactants to the MWCNTs solution to increase their dispersion in the polymer matrix. The ratio between Triton X100 and MWCNTs was 1:3.5. TangoPlus was then added to the solution and kept on a hot plate 80 °C with a magnetic stirrer at 400 RPM for 48 h. The solution was then further mixed in a high-speed mixer at 2500 RPM for an hour. The IL/polymer was then formed by mixing photo-curable mono-functional monomer, TangoPlus, and three wt.% of 1-ethyl-3-methylimidazolium tetrafluoroborate. Figure 12 [76] shows the schematic representation of the fabrication of the piezoresistive sensors. Followed by the casting and subsequent curing of photopolymer TangoPlus on the molds to develop an insulation layer, the first electrode layer was formed via a screen-printing technique. Six electrodes were formed using MWCNTs-polymer composites by putting a mask on the insulation layer. After the sample was cured at 80 °C for 5 min, the IL/polymer composite was poured on top of it. UV exposure was then done the sample to develop a piezoresistive layer having a thickness of 1 mm. Then, the second layer of electrodes was formed by the same technique to create two separate electrodes on the piezoresistive layer. Finally, TangoPlus was poured as the top layer and UV-exposed to form an insulation layer of 1 mm thickness. The sensor consisted of 12 taxels that included two layers of electrodes.

Another work on microcontact printing that used 3D-printed molds can be exemplified on the development of liquid-phase electronic circuits [77]. The elastomeric circuits were built with GaIn alloy via casting and needle injection techniques. A micro-patterned mold was developed using photolithography and cast on with silicone elastomer. The sample was cured and bonded to another elastomeric layer for sealing purposes. The embedded micro-channels inside the layer were then filled with liquid GaIn alloy using a needle and syringe. Finally, wires were externally connected to the terminals of the circuit, followed by sealing them with silicone elastomeric droplets. Park et al. [78] explained the fabrication of hyper-elastic pressure transducers with silicone rubber and liquid eutectic gallium–indium. The sensors were used for measuring pressure ranging between 0 and 100 kPa to determine the changes in resultant resistances. The dimensions of the fabricated microchannels were around 25 microns, developed using the mask-less soft lithographic technique. The soft lithography was done on PDMS after it was cast on SU-8 mold. The elastomer was removed from the mold after forming a hydrophobic monolayer via thermal deposition. The channels on these were formed by bonding the patterned PDMS to un-patterned PDMS using oxygen plasma treatment. Silicone rubber was cast and cured at room temperature for 4 h to form the micro-channels of the sensors. These elastomers were then peeled off and bonded on a thin uncured layer of EcoFlex. Then un-patterned elastomeric layers were spin-coated to form a thin layer on the micro-channels to avoid their exposure. The two elastomeric layers were then cured together to form proper bonding, followed by filling the channels with syringe-assisted GaIn. Finally, the channels were sealed with another layer of EcoFlex. Similar work was shown by King et al. [79] depicting the fabrication of bilayer arrays in milli-fluidic droplet traps formed from 3D printed molds. Figure 13 [79] shows the schematic diagram of the design of the developed chip. The presence of the inner droplet lipid bilayers along with the formation of the droplet arrays are shown in the diagram. The 3D printed molds with a thickness of 2 mm were developed using CAD and subsequently baked at 80 °C overnight to make them suitable for PDMS casting. The baking was done by attached the molds on a glass that was coated with non-polymerized PDMS. Then the molds were covered with a layer of trichloro (1H,1H,2H,2H-perfluorooctyl) silane via exposing them to silane vapor for an hour in desiccated conditions. The sample was then off the glass and punched to form access holes in them. The oxygen plasma treatment was done on the samples after they were again bonded to glass slides to seal the channels. Then, the channels were filled with a mixture of trichloro (1H,1H,2H,2H-perfluorooctyl) silane and ethanol and subsequently washed with ethanol. Finally, the samples were dried with nitrogen gas for 30 min and baked at 80 °C to ease the flow of the aqueous droplets through the channels.

Su et al. [80] used an inkjet printing technique to develop a 3D microfluidic device due to its high fabrication speed and good resolution. Figure 14a [80] shows the schematic diagram of the steps of fabrication using an inkjet technique to form the printed prototypes. The cross-sectional views as shown in Figure 14b,c [80] under different magnifications depict that the prototypes have high aspect ratios. Figure 14d [80] shows the sketch of the cross-sectional view of the prototypes to show its dimensional notions. PMMA was used to support the structure during the curing of the SU-8 ink. The printer can print the grid of 5 × 5 μm pixels which is a very minimal resolution for the pattern. For this microfluidic device, the ink used in this 3D printing process can be easily integrated with electronics which is the advantage of this process. The sensors fabricated by inkjet printing can fabricate low-cost, flexible electronics with a good frequency range. Dijkshoorn et al. [81] used thermoplastic polyurethane, polylactic acid (PLA) and conductive Inks in fused deposition modeling to develop a sensor for capacitive sensing and resistive sensing. The sensor is also capable of magnetic sensing and electrochemical sensing in their future applications. During the strain gauges and piezoresistive sensing, the sensor can be printed which allows conductive traces to be positioned ideally for the expected loading.

Table 2 shows a comparative study of certain parameters like reliability, scalability, and lifetime of the different molds developed for fabricating sensors for the above-mentioned industrial uses. It is seen that even though each of these molds has been employed to form sensors for a different purpose, they have served well to develop the sensors with high quality.

## 3. Current Challenges and Future Opportunities

Even though a lot of work has been done on 3D printed mold-based sensing, there are still some loopholes that need to be addressed in this sector. Firstly, this technique requires a 3D printer of high qualified to form molds of high quality. This makes it cost-ineffective. The cost of a 3D printer can only be dealt with by developing sensors on a large scale. Now, a large number of sensors are only designed and developed on an industrial scale, rather than a research-level. This can be dealt with redefining the attributes of the printer to form sensors with a high range of structural dimensions, the capability to be able to form different shapes of prototypes, able to handle a range of processing materials and working a dynamic range of temperature and humidity. Also, these printers can only deal with a few processing materials, especially that of printing filaments like ABS [82,83,84], PLA [85,86,87] and some polymers [88,89,90]. The range of printing filaments to form the molds should also be increased to develop molds with different layers, and subsequent casting and contact printing of the raw materials that can be processed within a single mold. Secondly, this technique does not allow sensors to be formed with a few micro and nano dimensions. The use of nanomaterials and nanoparticles with this technique has hardly been associated. This makes this technique to be less efficient, especially for other techniques like photolithography and other nanoelectromechanical systems-based techniques [91,92], which can generate high-quality nanostructure flexible sensors. The printing systems should be customized to deal with techniques like ink-jet printing and techniques handling as well as subsequently printing the nanomaterials to form the sensors. The formation of nanocomposite-based sensors [93,94,95,96] using this technique has drawbacks of electrical and mechanical characteristics in comparison to nanostructured-flexible sensors. One of the ways to deal with this problem is to familiarize the printing systems with nanoparticles to print them directly on the substrates. The casting method can be conjugated with other printing techniques to minimize the effect of surface tension, viscosity, and other contact parameters between two layers. Thirdly, the use of a lot of polymers is a restraint to be used with this technique. This is due to the adherence of polymers like PDMS [19,75,97,98] with the techniques of casting and microprinting techniques. The number of raw materials that can be associated with this technique should be increased to increase the versatility of the fabricated sensors. Table 3 summarizes some of the significant works done on the fabrication of 3D printed sensors using mold-based sensors with different types of materials and their advantages related to the chosen application.

A market survey has been done on the use of a 3D printing technique for fabricating sensing prototypes in recent years. It is seen that the use of 3D printed sensors has been around 2000 million ISD in the year 2017, along with an estimated growth in the rise of Compound Annual Growth Rate (CAGR) of 26.54% in the next five years [99,100]. Among the mentioned 3D printed sensors, MEMS sensors have the highest share, followed by image sensors. Some of the major applications where these sensors have been employed are automotive, healthcare, robotics, and defense. Among healthcare applications, these sensors have been implemented for viewing the different epidermal layers of skin, tumors, and veins. Based on the strengths of 3D printing, their presence would also be assisted by the Internet of Things (IoT) [101], where nanomaterials and conductive inks would be used to form the highly flexible, durable, and conductive sensing prototypes. An estimation of over more than 30 Billion USD has been done for the use of 3D printed sensors in the next few years [102]. This includes the use of a range of polymers and metals at different prices to be exploited with 3D printing techniques. These sensors have also been calculated to be used for aerospace, defense, medical, and healthcare applications. There is also an estimated rise in the compound annual rate to more than 16.5% by 2025, where different types of 3D printing will be utilized to develop sensing prototypes [103]. Figure 15 [103] shows the classification of different kinds of 3D printing techniques that have been preferred over the past few years to develop prototypes for different applications. Some of the common 3D printing techniques that would be popularized in the future years are photo-polymerization (SLA/DLP/CLIP), powder bed fusion (SLS/DMLS/EBM), material extrusion, material and binder jetting, directed energy deposition, and sheet lamination [104,105]. Few of the specific countries which would be having a higher market potential for 3D printing are United States, Canada, Mexico, Japan, China, India, France, Germany, United Kingdom, Italy, and Spain [106]. Along with the emphasis given on the strategies and collaborations in regards to the fabrication technique and finished product, the impact analysis would also be done on the potential for commercialization on the technological and economic trends. The continuous demand for 3D printed sensors has been mostly from the consumer electronics, automotive and building automation industries. The PR news wire reviewed that the projected rise of global printed sensors would be over 12 billion dollars by 2022 [107]. Even though this includes all the sectors where the sensors have been employed, biomedical sensors account for the largest share in the past as well as forecasted period. The real-time measurement of the physiological parameters of a patient like glucose level and blood pressure has led to an increase in the demand of 3D printed sensors. This has led to an increase in the operational efficiency and productivity of the sensors. It has also been due to the increased human-machine interfaces that the 3D printed sensors having simple structure and operating principle have been largely popularized. Some of the types of biomedical sensing prototypes that has been emphasized on are capacitive, piezoelectric, piezoresistive and optical sensors.

## 4. Conclusions

This presents an overview of some of the significant works done on the design and fabrication of 3D printed mold-based sensing prototypes. The design was done using a range of drawing software associated with the 3D printing device. Then, different kinds of 3D printing techniques were used to form the prototypes with dimensions suitable for a particular application. Some of the advantages of these techniques the customization of the designs, quick fabrication of the prototypes on a large scale, the formation of less electronic waste, the dynamic range of fabricated products, and the easy accessibility of the device. Even though some of the drawbacks that are mentioned above need to be dealt with, the use of the combination of molding and casting techniques of 3D printing is a highly popular technique among different research groups in the world. This technique allows for producing prototypes for different applications. The next steps would be to deal with the drawbacks as well as a conjugate with this technique with other sensor fabrication techniques to fabricate sensors with high quality in terms of longevity and sensitivity. If this technique is used effectively as one of the primary fabrication methods to develop sensor prototypes, a range of devices can be developed for monitoring purposes.

## Figures and Tables

**Figure 1 sensors-20-00703-f001:**
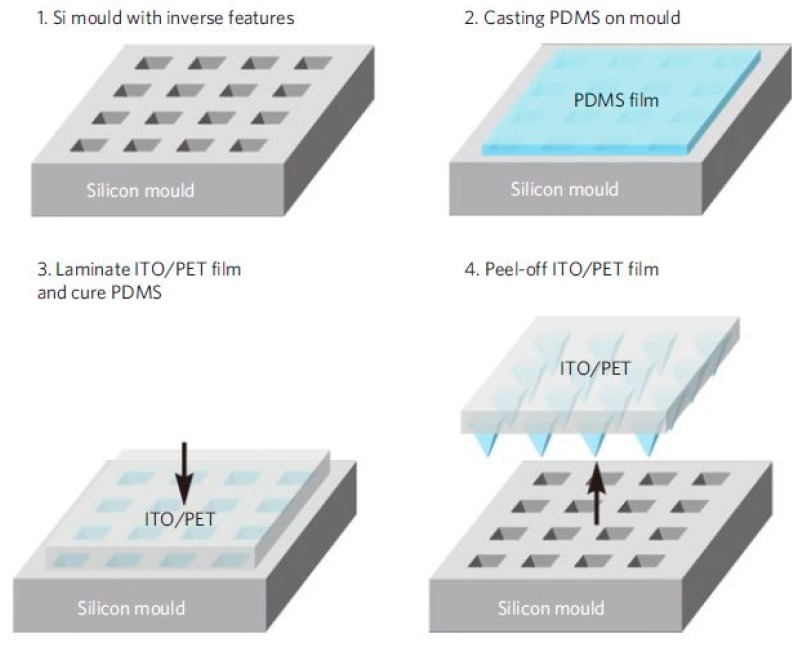
Schematic diagram of the steps of the fabrication process of the microstructure Polydimethylsiloxane (PDMS) films [47]. Reproduced from Mannsfeld, S.C.; Tee, B.C.; Stoltenberg, R.M.; Chen, C.V.H.; Barman, S.; Muir, B.V.; Sokolov, A.N.; Reese, C.; Bao, Z. Highly sensitive flexible pressure sensors with microstructured rubber dielectric layers. *Nat. Mater.*
**2010**, *9*, 859.

**Figure 2 sensors-20-00703-f002:**
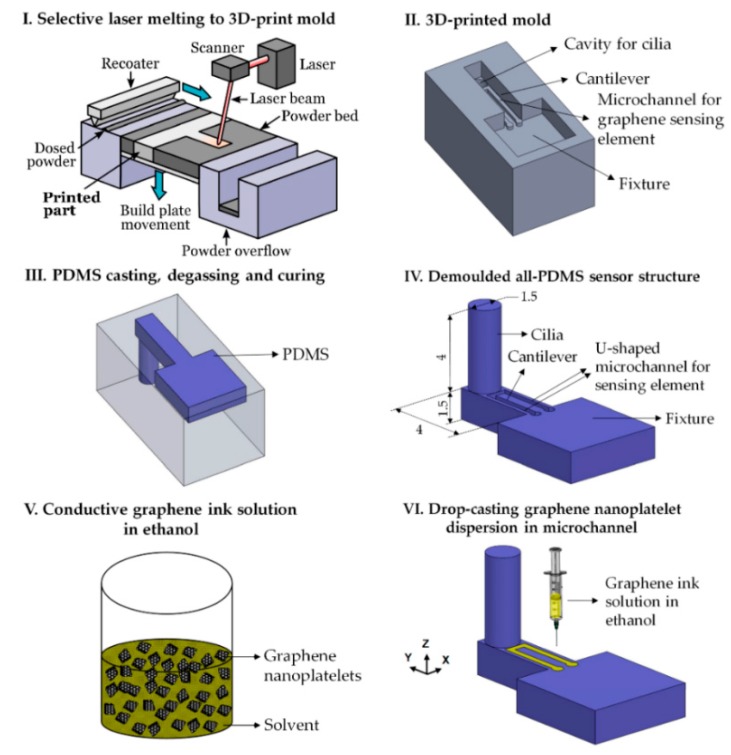
Schematic diagram of the fabrication process of the 3D printed molds and subsequent casting of the PDMS and graphene on the mold and microchannel, respectively [50]. Reproduced from Kamat, A.M.; Pei, Y.; Kottapalli, A.G. Bioinspired cilia sensors with graphene sensing elements fabricated using 3D printing and casting. *Nanomaterials*
**2019**, *9*, 954.

**Figure 3 sensors-20-00703-f003:**
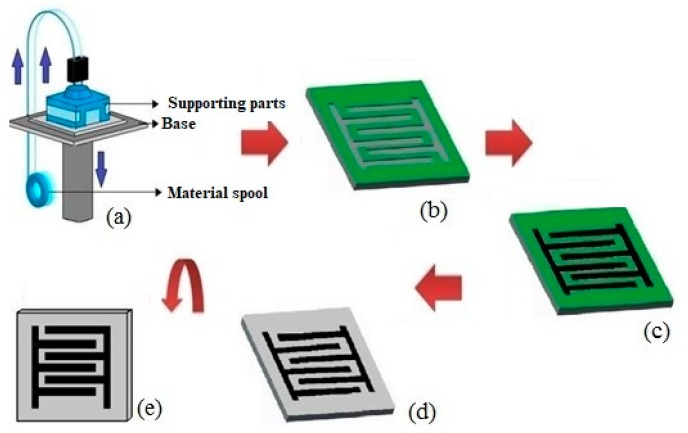
Illustration of the bending flexibility of the Graphite-Polydimethylsiloxane (PI) sensor patch. (**a**) 3D printing was done to develop (**b**) the molds on top on which (**c**) graphite and PDMS were cast to form the electrodes and substrates, respectively. (**d**) The sample was desiccated, cured, and peeled off to form (**e**) the sensor patches. [35]. Reproduced from Nag, A.; Feng, S.; Mukhopadhyay, S.C.; Kosel, J.; Inglis, D. 3D printed mold-based graphite/PDMS sensor for low-force applications. *Sens. Actuators A Phys.*
**2018**, *280*, 525–534.

**Figure 4 sensors-20-00703-f004:**
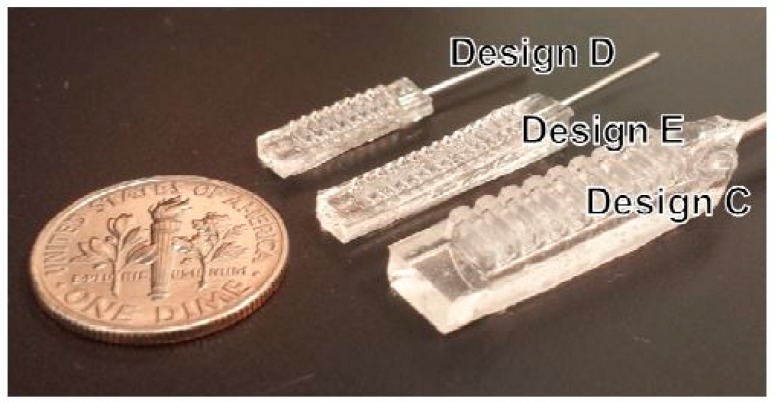
Representation of the formed balloon fin actuators in different geometries [55]. Reproduced from Hwang, Y.; Paydar, O.H.; Candler, R.N. Pneumatic microfinger with balloon fins for linear motion using 3D printed molds. *Sens. Actuators A Phys.*
**2015**, *234*, 65–71.

**Figure 5 sensors-20-00703-f005:**
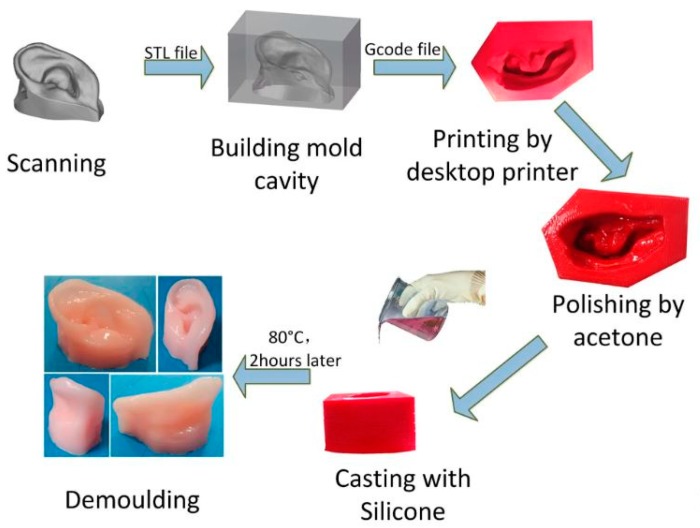
Representation of the individual steps of the molding and demolding processes to form soft tissue prostheses [57]. Reproduced from He, Y.; Xue, G.H.; Fu, J.Z. Fabrication of low cost soft tissue prostheses with the desktop 3D printer. *Sci. Rep.*
**2014**, *4*, 6973.

**Figure 6 sensors-20-00703-f006:**
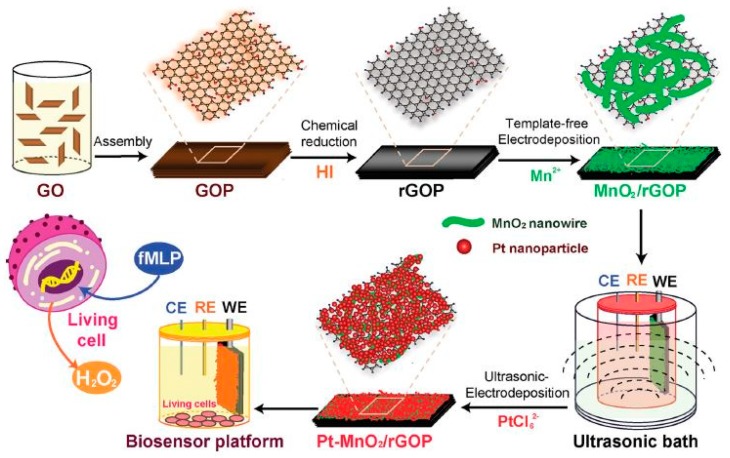
Schematic diagram of the fabrication process of the paper-based electrodes [59]. Reproduced from Xiao, F.; Li, Y.; Zan, X.; Liao, K.; Xu, R.; Duan, H. Growth of metal–metal oxide nanostructures on freestanding graphene paper for flexible biosensors. *Adv. Funct. Mater.*
**2012**, *22*, 2487–2494.

**Figure 7 sensors-20-00703-f007:**
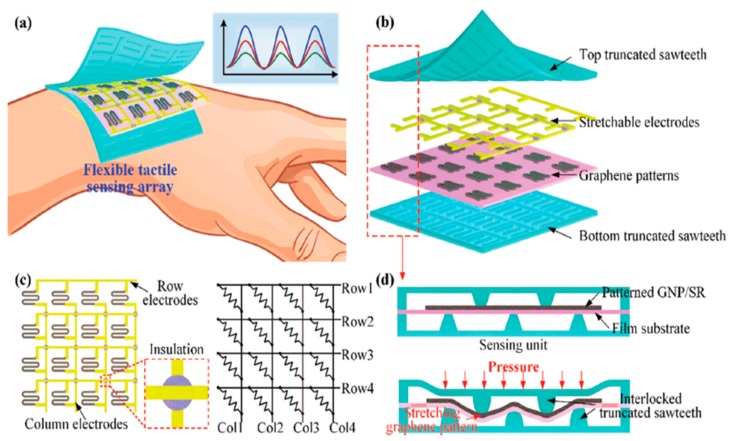
(**a**) Schematic diagram of the highly flexible tactile sensor depicting an interlocked truncated saw-tooth structure; (**b**) structural design of the tactile sensor; (**c**) electrode configuration of the sensing device; (**d**) the cross-sectional view of the sensing unit for pressure detection. [60]. Reproduced from Wang, Y.; Zhu, L.; Mei, D.; Zhu, W. Highly flexible tactile sensor with an interlocked truncated sawtooth structure based on stretchable graphene/silver/silicone rubber composites. *J. Mater. Chem. C*
**2019**.

**Figure 8 sensors-20-00703-f008:**
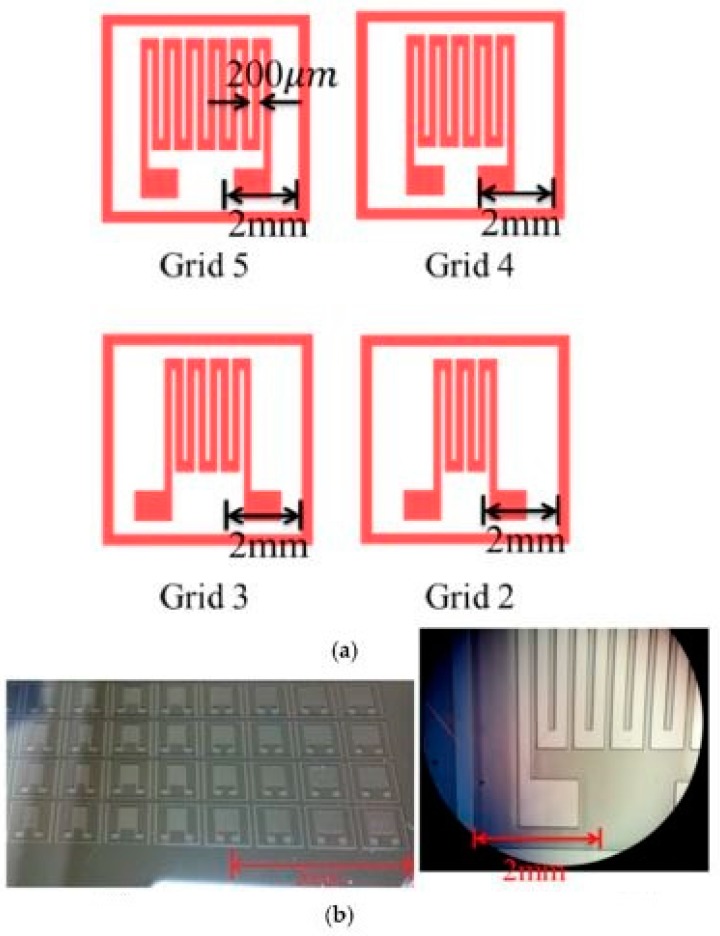
(**a**) Schematic diagram of the patterns for the molds for sensor fabrication. (**b**) Optical microscopic image of the molds [65]. Reproduced from Han, C.J.; Chiang, H.P.; Cheng, Y.C. Using micro-molding and stamping to fabricate conductive polydimethylsiloxane-based flexible high-sensitivity strain gauges. *Sensors*
**2018**, *18*, 618.

**Figure 9 sensors-20-00703-f009:**
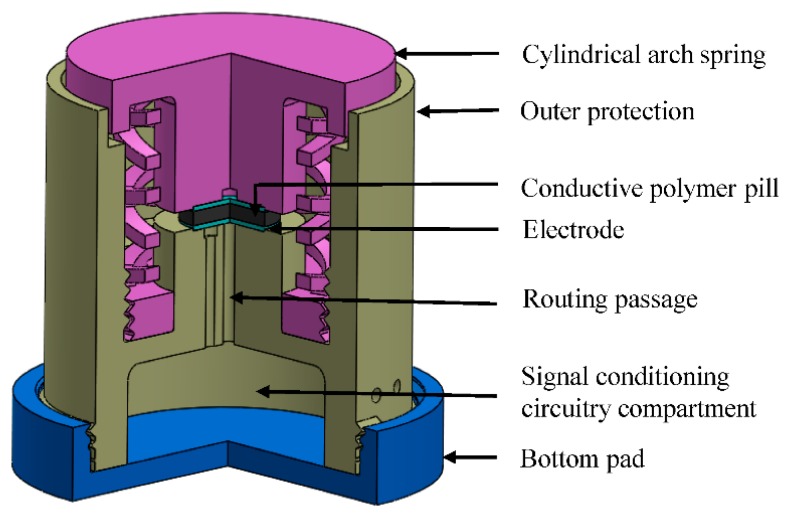
Schematic diagram of the three-quarter view sector of the developed sensor [66]. Sampath, P.; De Silva, E.; Sameera, L.; Udayanga, I.; Amarasinghe, R.; Weragoda, S.; Mitani, A. Development of a conductive polymer-based novel 1- Degree of Freedom (DOF) tactile sensor with cylindrical arch spring structure using 3D printing technology. *Sensors*
**2019**, *19*, 318.

**Figure 10 sensors-20-00703-f010:**
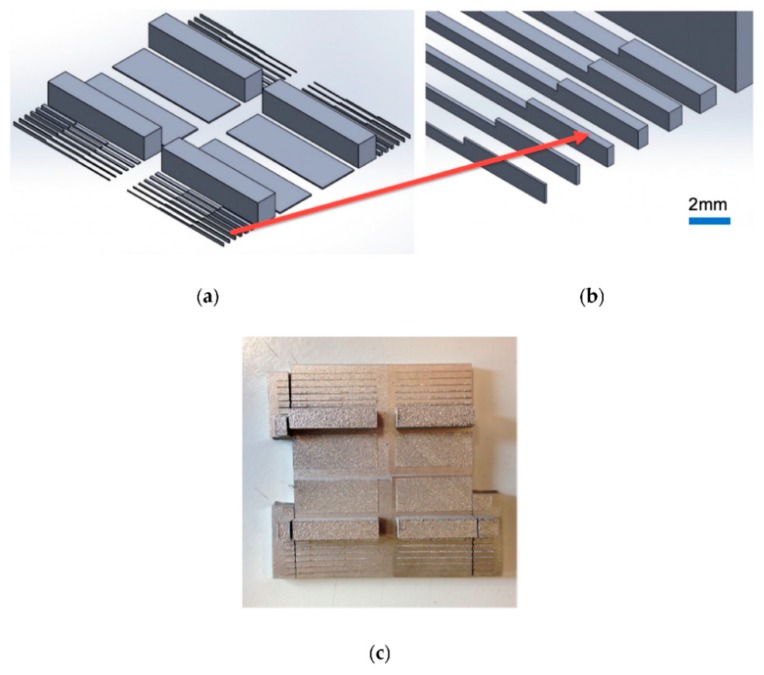
(**a**) Arrays of micro ridges that were printed using selective laser sintering. (**b**) Enlarged view of the rectangular patters. (**c**) Printed patterns on the roughness plates [69]. Reproduced from Zhang, N.; Liu, J.; Zhang, H.; Kent, N.J.; Diamond, D.; Gilchrist, M.D. 3D printing of metallic microstructured mold using selective laser melting for injection molding of plastic microfluidic devices. *Micromachines*
**2019**, *10*, 595.

**Figure 11 sensors-20-00703-f011:**
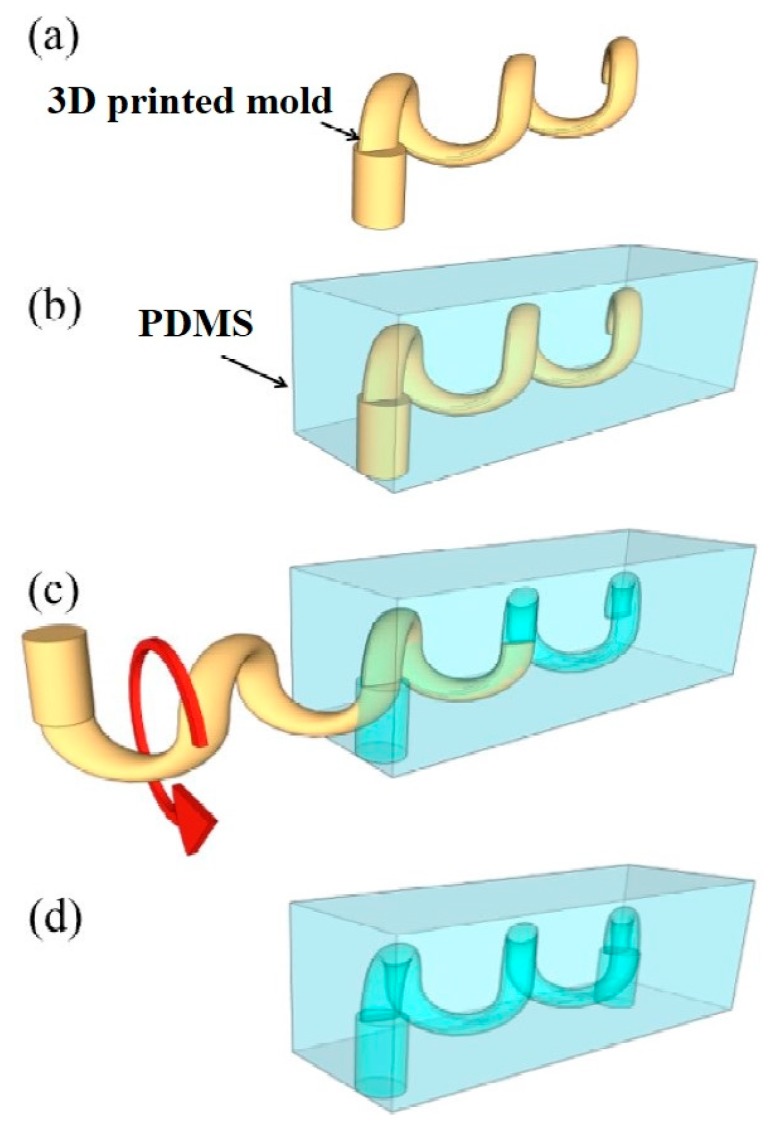
The fabrication process of the (**a**) 3D printed mold. (**b**) Pouring and curing of the pre-polymer to form the replica mold. (**c**) Extraction of the mold and subsequently rotated. (**d**) Finish product depicting the non-planar PDMS microfluidic channel. Reproduced from Hwang, Y.; Paydar, O.H.; Candler, R.N. 3D printed molds for non-planar PDMS microfluidic channels. *Sens. Actuators A Phys.*
**2015**, *226*, 137–142 [75].

**Figure 12 sensors-20-00703-f012:**
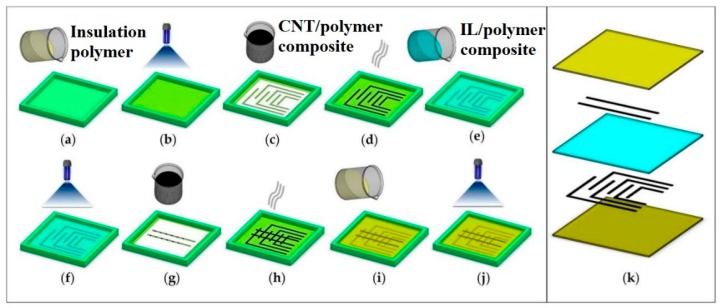
Schematic diagram of the fabrication of piezoresistive sensors using multi-walled carbon nanotubes (MWCNTs)-polymer composites. (**a**), (**b**) TangoPlus is poured onto the formed mold and cured to form the insulation layer; (**c**), (**d**) The screen printing and thermal curing of the MWCNTs/polymer paste were done to form the first electrode layer; (**e**), (**f**) A composite formed by ionic liquid and polymer was poured on the mold and cured to form a piezoresistive intermediate layer; (**g**), (**h**) The second layer was then formed by pouring and curing of the MWCNTs/polymer paste; (**i**), (**j**) Finally, TangoPlus was poured and subsequently cured onto the samples to form the top insulation layer; (**k**) Enlarged view of the sensor. Reproduced from Emon, M.; Choi, J.W. Flexible piezoresistive sensors embedded in 3D printed tires. *Sensors*
**2017**, *17*, 656. Reproduced from Emon, M.; Choi, J.W. Flexible piezoresistive sensors embedded in 3D printed tires. *Sensors*
**2017**, *17*, 656 [76].

**Figure 13 sensors-20-00703-f013:**
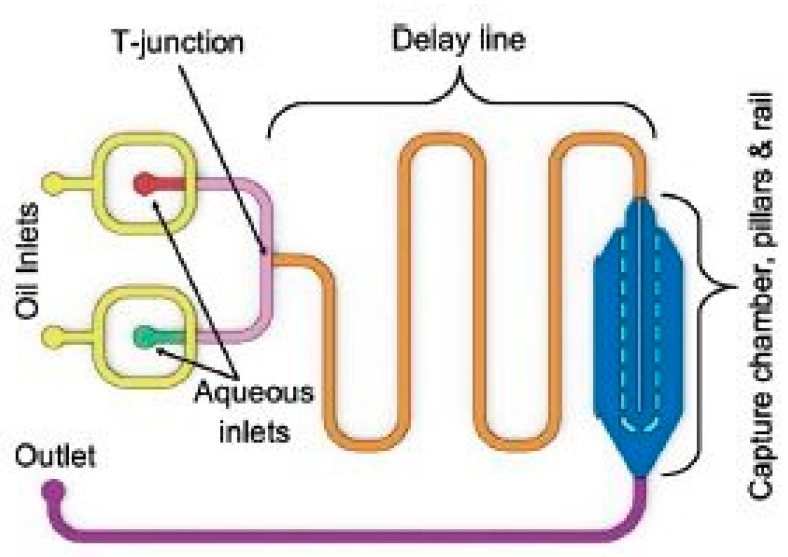
Schematic diagram of the fabrication of the microfluidic chip. The different colors in the figure indicate the individual components of the chip [79]. Reproduced from King, P.H.; Jones, G.; Morgan, H.; de Planque, M.R.; Zauner, K.P. Interdroplet bilayer arrays in millifluidic droplet traps from 3D-printed molds. *Lab Chip*
**2014**, *14*, 722–729.

**Figure 14 sensors-20-00703-f014:**
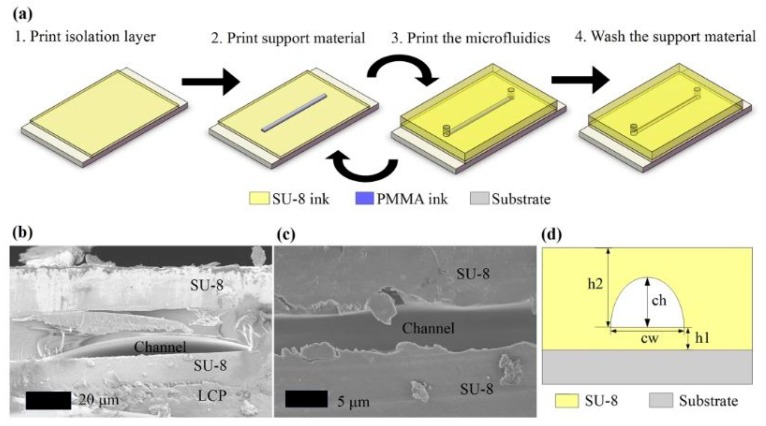
Schematic diagram of the steps of fabrication for the development of the inkjet-utilized printed prototypes. **(a)** Schematic diagram of the steps of fabrication for the development of the inkjet-utilized printed prototypes; (**b**) Cross-sectional views show that the sensors have aspect ratios, as given in with; (**c**) different magnifications; (**d**) A sketch to showcase the cross-sectional view with dimensional notations [80]. Reproduced from Su, W.; Cook, B.S.; Fang, Y.; Tentzeris, M.M. Fully inkjet-printed microfluidics: a solution to low-cost rapid three-dimensional microfluidics fabrication with numerous electrical and sensing applications. *Sci. Rep.*
**2016**, *6*, 35111.

**Figure 15 sensors-20-00703-f015:**
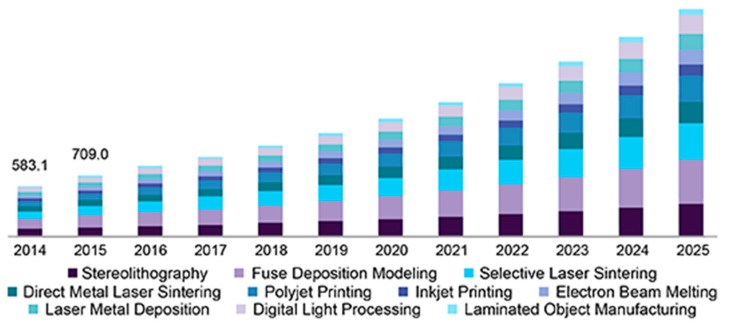
Representation of the use of different 3D printing techniques over the years for developing sensing prototypes [104]. Reproduced from Available online: https://www.grandviewresearch.com/industry-analysis/3d-printing-industry-analysis.

**Table 1 sensors-20-00703-t001:** Comparative study of some of the significant works done on 3D printed mold-based sensing systems.

Electrode Material	Substrate Material	Advantages	Application	Ref.
Graphite	PDMS	Dual Characteristics for electrochemical and strain sensing	Low-force sensing, tactile sensing	[35]
Gold, Ag/AgCl	PDMS	Probability and easy-to-use.	Cell and tissue diagnostics	[27]
Silver nanowires	PDMS	High gauge factor, excellent piezoresistivity.	Strain sensing	[36]
Single-Walled Carbon Nanotubes (SWCNTs)	PDMS	Transparent and stretchable nature. High conductivity in a stretched state.	Tactile sensing	[37]
Hydrogel	Polyethylene	Biocompatible with human tissue Thermo-resistive in nature	Tactile sensing, electronic skins	[38]
Silver trifluoroacetate	poly(styrene-*b*-butadiene-*b*-styrene)	excellent stretchability, strong interfacial connections between conductive patterns and substrate	Vibrational sensing	[39]
Multi-Walled Carbon Nanotubes (MWCNTs)	PDMS	High stretchability (300%) High resistance stability	Wearable electronic devices	[40]
Conductive fluid	Silicone elastomer Dragonskin 10	High accuracy in response Quick response	Strain sensing	[41]
Copper, Polyethylene terethaphlate, Indium titanium oxide	PDMS	Very low detection limit Fast response and recovery time Excellent durability Good tolerance to variations to ambient temperature and humidity	Human physiological signals, pressure sensing of wrist pulse	[42]
Carbon fiber	PDMS	Thermal stability, superior mechanical strength Possibility of a roll-to-roll process Low capacitive current	4-point sensing measurement	[43]

**Table 2 sensors-20-00703-t002:** Representation of some of the attributes of some of the significant sensors mentioned above for industrial uses.

Sensor Materials	Reliability	Scalability of the Mold(Minimum)	Lifetime	Ref.
Carbon particles, PDMS	High	200 microns	Long lifetime because of the use of the photolithography technique to form the molds. Can withstand over sixty stretch-release cycles.	[65]
Carbon black, silicon dioxide	High reliability and repeatability in response with a maximum standard deviation of 0.157 V.	10 mm	Can withstand over then loading and unloading test with a maximum standard deviation in response of 0.117 mm.	[66]
Stainless-steel CL 201ES powder	High reliability in the fabrication due to selective laser sintering process.	0.1 mm	High lifetime due to the use of mold release agent to deal with demolding difficulties and surface roughness.	[69]
PDMS	Medium, due to surface roughness and replication fidelity.	100 microns	Long lifetime because of the use of soft lithography techniques to develop the microchannel.	[75]
MWCNTs, photo-polymer	Medium lifetime due to the restrictions of the wires from the circuit board during the movement of the car.	80 mm	High lifetime due to the use of a direct-print polymerization process for rectifying the limitation of manufacturing scalability and design flexibility.	[76]

**Table 3 sensors-20-00703-t003:** Comparison between some of the significant works done on mold-based 3D printing for a range of materials and their associated advantages for the chosen application.

Sensor Materials	Application	Advantages	Limitation	Ref.
Graphite, PDMS	Low force sensing, phosphate sensing	Highly flexible, capacitive, multifunctional in nature	Not very small in size, not selective in nature.	[35]
2-[[(Butylamino)carbonyl] oxy]ethyl acrylate, 1-ethyl-3-methyl-imidazolium tetrafluoroborate (EMIMBF4), MWCNTs	Tactile sensing	High sensitivity, high electrical conductivity	Variation of sensitivity with limited mobility of polymer chains and ion liquid domain.	[49]
Graphene, PDMS	Flow sensing	High resolution, high aspect ratio, high gauge factor, high sensitivity	Not small in size, high wt.% of graphene required to form the sensor.	[50]
PDMS	Reduction of stress concentrations	The capability of multi-directional object transfer,	Limited miniaturization of the prototype.	[55]
Reduced graphene oxide, manganese oxide, platinum	Detection of extracellular H_2_O_2_ from human liver cancer cells	Broad linear dynamic range, high sensitivity, selectivity, long stability, good reproducibility	Difficult to take off the freestanding nanoparticles from the template.	[59]
Graphene, silicone rubber	Tactile sensing	High electrical conductivity, high gauge factor	High sensitivity for low force.	[60]
Carbon particles, PDMS	Strain sensing	High piezoresistive coefficient	Non-uniform gauge factor.	[65]
Stainless-steel CL 20ES powder	Monitor nitrate concentrations	High flatness of the prototypes, cost-effective, fast prototyping, flexibility in the design	Limited precision and surface finish.	[69]
MWCNTs, photopolymer	Piezoresistive sensing in 3D printed tires	High electrical conductivity and flexibility, high sensitivity	Restriction of the movement of the car due to the intertwining of wires of the circuit board.	[76]
PDMS	Transfer of osmotic water	High reproducibility, combined electrical and optical access	Requirement of 48 h to form a device-to-device cycle, cannot be used for a high amount of droplets.	[79]

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
