# Peer review of "Recent Progress in 3D Printed Mold-Based Sensors"

_sensors, 2020, doi:10.3390/s20030703_

Round 1

Reviewer 1 Report

This manuscript reports the recent progress in 3D Printed Mold-based sensors. The results are very helpful to researchers in 3D printed mold field and relative fields. The manuscript is organized with a brief introduction, the fabrication procedure of the molds and their application.

In each topic, and to help the readers of this work, I expected to see some visual examples (figures) with the main results described. The readers should be able to assess the quality and relevance of the presented works when they are reading this paper. It is not hands-on when we read a review and we need to search other papers to understand what the Autor’s are describing.

Manuscript sections should be rearranged as topic and subtopic of each mold application, such as:

3D printed mold-based sensors for different applications

2.1Biomedical

2.2 Industrial

2.3 Environmental uses.

In the text of the manuscript, please define acronyms or abbreviations or variables when first used and only once. However, do not define acronyms or abbreviations or variables if they are not used in the text. Make sure that acronyms or abbreviations or variables that are not reused in the text and captions are spelled out.

 In the text of the manuscript, please place the figures when first cited. For example, Figure 1 was cited in the line 152 and the figure appear in the line 183, in another paragraph.

In some figures, the characters are too small.  

In the text of the manuscript use the same spelling. The word mold and mould appear a lot of times, please be coherent.

As a report in recent progress in 3D Printed Mold-based sensors, many of the references (~25%) are too old.

Reviewer 2 Report

The paper presents a review of different works developed in the field of molding fabrication technologies applied to sensor and actuator applications.

The manuscript is sufficiently informative and well organised overall.

It lacks coverage of aspects related to reliability, scalability and lifetime of the different molds for industrial use of the fabrication routes proposed in the cited works.

The market analysis presented in the abstract and conclusions is merely reporting some generic data of commercial market analysis reports. As such the significance  of this analysis is pretty limited and does not contribute sufficiently to the scope and aim of this review.  

Reviewer 3 Report

Authors have written an excellent overview of current research on design and fabrication of the different 3D printed molds.

Line 60-62 – Authors state that no current reviews mention a particular 3D printing type or application. Author needs to do a more comprehensive literature search as such reviews have been published. Authors explain the printing and mold formation in detail for each paper they cite. Perhaps, the authors can consider summarizing these details more and focusing on the advantages of using these different materials for 3D-printing or molds and how they aided in the different applications. Authors can explain in further detail what some of the biomedical and environmental applications are, example line 278. How successful were these applications? How have they pushed the field forward?

Round 2

Reviewer 1 Report

Please use the word mold instead mould.